# The PEMDAC phase 2 study of pembrolizumab and entinostat in patients with metastatic uveal melanoma

Lars Ny[1,12 ✉], Henrik Jespersen [1,2,12], Joakim Karlsson[3,4], Samuel Alsén[3], Stefan Filges [5], Charlotta All-Eriksson[6], Bengt Andersson[7], Ana Carneiro[8], Hildur Helgadottir[9], Max Levin[1], Ingrid Ljuslinder[10], Roger Olofsson Bagge[3], Vasu R. Sah[3], Ulrika Stierner[1], Anders Ståhlberg [5], Gustav Ullenhag[11], Lisa M. Nilsson[3,4] & Jonas A. Nilsson [3,4 ✉]

Preclinical studies have suggested that epigenetic therapy could enhance immunogenicity of cancer cells. We report the results of the PEMDAC phase 2 clinical trial ($n = 29$; NCT02697630) where the HDAC inhibitor entinostat was combined with the PD-1 inhibitor pembrolizumab in patients with metastatic uveal melanoma (UM). The primary endpoint was objective response rate (ORR), and was met with an ORR of 14%. The clinical benefit rate at 18 weeks was 28%, median progression free survival was 2.1 months and the median overall survival was 13.4 months. Toxicities were manageable, and there were no treatment-related deaths. Objective responses and/or prolonged survival were seen in patients with *BAP1* wildtype tumors, and in one patient with an iris melanoma that exhibited a UV signature. Longer survival also correlated with low baseline ctDNA levels or LDH. In conclusion, HDAC inhibition and anti-PD1 immunotherapy results in durable responses in a subset of patients with metastatic UM.

**Trial registration** ClinicalTrials.gov registration number: NCT02697630 (registered 3 March 2016). EudraCT registration number: 2016–002114-50.

[1] Sahlgrenska Cancer Center, Department of Oncology, Institute of Clinical Sciences, University of Gothenburg and Sahlgrenska University Hospital, Gothenburg, Sweden. [2] Department of Oncology, Akershus University Hospital, Lørenskog, Norway. [3] Sahlgrenska Cancer Center, Department of Surgery, Institute of Clinical Sciences, University of Gothenburg and Sahlgrenska University Hospital, Gothenburg, Sweden. [4] Harry Perkins Institute of Medical Research, University of Western Australia, Perth, WA, Australia. [5] Department of Laboratory Medicine, Wallenberg Centre for Molecular and Translational Medicine, Department of Clinical Genetics and Genomics, Sahlgrenska Cancer Center, Institute of Biomedicine, University of Gothenburg and Sahlgrenska University Hospital, Gothenburg, Sweden. [6] Department of Oncology, St. Erik Eye Hospital, Stockholm, Sweden. [7] Department of Clinical Immunology and Transfusion Medicine, Sahlgrenska University Hospital, Gothenburg, Sweden. [8] Department of Hematology Oncology and Radiation Physics, Skåne University Hospital, and Institute of Clinical Sciences, Lund University, Lund, Sweden. [9] Present address: Department of Oncology, Karolinska University Hospital, Stockholm, Sweden. [10] Department of Oncology, Norrlands University Hospital, Umeå, Sweden. [11] Department of Oncology, Uppsala University Hospital, Uppsala, Sweden. [12] These authors contributed equally: Lars Ny, Henrik Jespersen. ✉email: lars.ny@oncology.gu.se; jonas.nilsson@perkins.org.au

Uveal melanoma (UM) is a rare form of melanoma, with an incidence of approximately eight new cases per million per year in Sweden[1]. UMs originate from choroid, ciliary body, or iris melanocytes and are clinically and biologically different to cutaneous melanoma[2,3]. The primary disease can in most cases be successfully treated with radiotherapy or enucleation, but almost one half of patients subsequently develop metastatic disease, usually to the liver[4,5]. While targeted therapies and immune-checkpoint inhibitors have revolutionized the treatment of metastatic cutaneous melanoma[6–8], there are still no effective treatments for patients with metastatic UM, who have a median survival of less than 12 months with the current available therapies[9].

In contrast to the *BRAF*, *NRAS*, or *NF1* mutations commonly found in cutaneous melanomas, metastatic UMs frequently harbor oncogenic mutations in the genes encoding G-protein-alpha protein *GNAQ* or the mutually exclusive *GNA11*, *PLCB4*, or *CYSLTR2*, most often together with monosomy of chromosome 3 (Chr. 3) and inactivating mutations of the *BAP1* tumor suppressor gene[10–13]. UM appears to show some immune responsiveness, since expanded and adoptively transferred tumor-infiltrating lymphocytes (TILs) have therapeutic clinical effects[13,14]. Tebentafusp, a bispecific protein immunotherapy targeting CD3 and melanoma-specific gp100, has shown very promising activity in the subset of patients with a HLA-A2 genotype[15], but outcomes with immune-checkpoint inhibitor monotherapy have been disappointing, with response rates typically below 5%[16,17]. Combined PD-1 and CTLA4 immune-checkpoint inhibition appears to be more effective but not as effective as in cutaneous melanoma[18,19].

Poor responses to checkpoint inhibitors are multifactorial and include low tumor mutational burden (TMB)[20], poor antigen processing and presentation, and immune-suppressive tumor microenvironments[21–23]. Drugs targeting epigenetic regulators such as histone deacetylases (HDACs) show promise as cancer therapies by reversing oncogene transcription and modifying the tumor microenvironment[24]. For instance, HDAC inhibitors can increase immunogenicity through several mechanisms such as blocking the effects of myeloid-derived suppressor cells (MDSCs) and regulatory T cells (Tregs)[25,26]; enhancing the expression of cancer antigens silenced by immunoediting[27]; and/or triggering DNA damage and cell death to activate danger signals and recruit immune cells[28,29]. Furthermore, HDAC inhibitors increase HLA class I expression in several cancer types, including UM[30,31].

However, we and others have also shown that HDAC inhibitors induce PD-L1 to inactivate T cells[31,32]. Nuclear acetylated PD-L1 was also recently shown to stimulate antigen presentation[33], providing a potential explanation for why PD-L1-high tumors are sensitive to PD-1 inhibition. These data suggest that anti-PD-1 therapies and HDAC inhibitors could synergize. Indeed, in vivo preclinical studies[26,31,34,35] and ongoing phase I/II trials have shown encouraging results when combining the class I HDAC inhibitor entinostat with the PD-1 inhibitor pembrolizumab in patients with PD-1 inhibitor-refractory cutaneous melanoma or lung cancer[36,37]. Also, other HDAC inhibitors synergize with PD-1 inhibitors in animal models[32,38,39], and combined vorinostat and pembrolizumab is clinically active in lung or head and neck cancer patients[40,41]. However, it is unknown whether this combination is effective in melanomas not harboring the usual mutations in *BRAF*, *NRAS*, or *NF1* and with a low TMB, such as UM.

Metastatic UM is a life-threating condition and, given the lack of established and effective treatments, new therapeutic strategies are urgently required. Here we conduct a clinical trial (the multicenter phase II PEMDAC study) testing the hypothesis that HDAC inhibition with entinostat increases UM immunogenicity

| Table 1 Baseline patient characteristics (n = 29 patients). | |
| --- | --- |
| **Characteristic** | |
| Age, median (range) | 70 (34–83) |
| *Gender, n (%)* | |
| Female | 12 (41) |
| Male | 17 (59) |
| *ECOG PS, n (%)* | |
| 0 | 24 (83) |
| 1 | 5 (17) |
| *Previous treatment for metastatic disease, n (%)* | |
| Surgery | 5 (17) |
| Radiotherapy | 1 (3) |
| Chemotherapy | 8 (28) |
| Isolated hepatic perfusion | 8 (28) |
| No previous treatment | 12 (41) |
| Time from metastatic disease until first dose of study drug (months), median (range) | 6.8 (0.5–53.6) |
| *Metastatic stage (AJCC 8th edition), n (%)* | |
| M1a | 17 (59) |
| M1b | 8 (28) |
| M1c | 3 (10) |
| N/A | 1 (3) |
| *Metastatic sites, n (%)* | |
| Liver only | 10 (34) |
| Extrahepatic only | 3 (10) |
| Liver and extrahepatic | 16 (55) |
| LDH > ULN, n (%) | 14 (48) |

Abbreviations: ECOG PS Eastern Cooperative Oncology Group performance status, LDH lactate dehydrogenase, ULN upper limit of normal

and responses to immune-checkpoint inhibition with pembrolizumab. In doing so, we provide evidence that a small subset of patients benefits from combined epigenetic therapy and immunotherapy. Among responders were patients with a *BAP1* wild-type status and a patient with a very rare form of uveal melanoma originating in the iris.

## Results

**Patient characteristics**. Twenty-nine patients were enrolled between February and December 2018, with a cutoff for the present analysis in December 2019, i.e., 12 months after the last enrolled patient received the first dose. The median follow-up for overall survival (OS) was 14.8 months. The study is ongoing, and the data presented here are a prespecified mature analysis of the primary endpoint of objective-response rate (ORR). Secondary endpoints and where they are reported can be found in Supplemental Information. Analyses are based on all patients who received at least one dose of study drug.

Patient characteristics at baseline are shown in Table 1. The median age was 70 years (range, 34–83 years), and 90% had liver metastases. Twelve patients (41%) had received no previous treatment for metastatic disease and eight patients (28%) had received previous chemotherapy for UM.

**Efficacy**. Treatment-response characteristics as per RECIST v1.1 criteria are shown in Fig. 1a–c, Supplementary Fig. 1a, b, and Supplementary Data 1. Twenty-eight patients had at least one follow-up radiological evaluation. One patient was excluded due to a protocol violation (did not fulfill RECIST criteria for radiological evaluation) in the first week following the first dose. Partial response (PR) was confirmed in four patients (one of whom was in the first cohort of ten patients), giving an ORR of 14% (95% CI, 3.9–31.7). The trial therefore met its primary endpoint. Eight patients (28%) showed clinical benefit (PR or stable disease, SD) at 18 weeks. Three out of four responses were

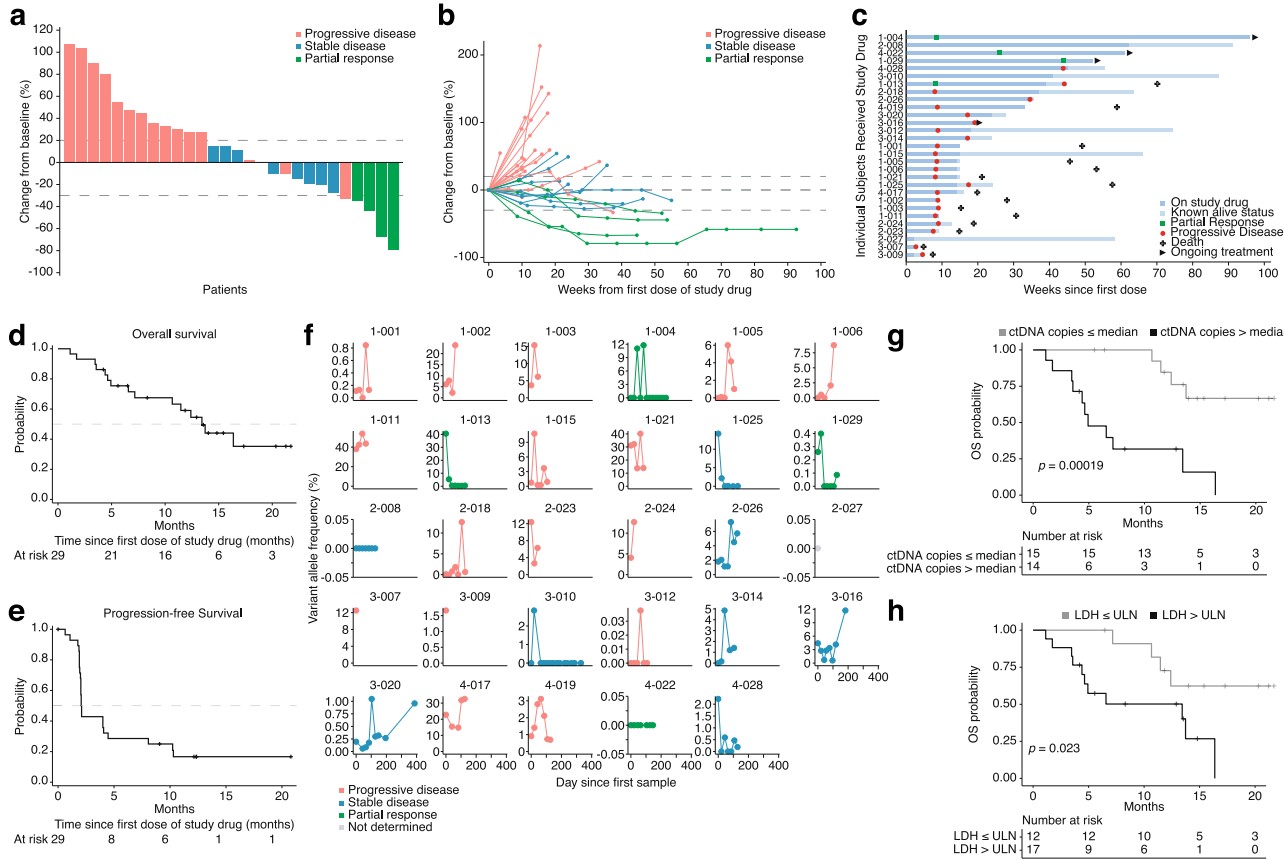

**Fig. 1 Clinical outcome data from the PEMDAC trial at data cutoff in December 2019. a** Waterfall plot showing maximum change in sum of target lesion diameter from baseline to data cutoff. One patient did not have a response assessment after baseline and is not included in the figure. Dotted lines represent thresholds for progressive disease (+20%) and partial response (−30%), respectively, according to RECIST v1.1 criteria. **b** Spider plot showing change in the sum of target lesion diameter over time for all patients with at least one follow-up scan. **c** Swimmer's plot showing time on treatment, time to best response, and duration of response in all patients who received at least one dose of study drug. In panels (a,b), *n* = 28 and in (c) *n* = 29 patients are shown, respectively. **d** Kaplan–Meier analysis showing overall survival (OS) of all patients. **e** Kaplan–Meier analysis showing progression-free survival (PFS) of all patients except one. **f** Circulating tumor DNA levels in patients with UM. Plasma from twenty-five patients was analyzed for the presence of mutated reads in either *GNA11* (Q209L) or *GNAQ* (Q209L/P). Variant allele frequencies (VAF) in percent compared with reads with wildtype alleles are plotted. **g** Kaplan–Meier analysis comparing OS between patients with high or low (relative to median) levels of total detected ctDNA copies, *n* = 14 and *n* = 15 patients, respectively. **h** Kaplan–Meier analysis comparing OS between patients with lactate dehydrogenase (LDH) baseline greater or lower than the upper limit of normal (ULN), *n* = 17 and *n* = 12 patients, respectively. In (g, h), *p*-values for survival associations were calculated using log-rank tests. No adjustments for multiple comparisons were made. All statistical tests were two-sided. Source data are provided as a Source Data file.

ongoing at data cutoff. Median overall survival (OS) was 13.4 months and OS at one year was 59%. Median progression-free survival (PFS) was 2.1 months and one-year PFS was 17% (Fig. 1d, e). Levels of circulating tumor DNA (ctDNA) were measured using NGS from before treatment start and after several rounds of treatment. The pattern was complex with spikes of ctDNA appearing in many of the patients' blood (Fig. 1f, Supplementary Data 2), possibly resulting from waves of proliferation. Twelve of sixteen patients with progression of disease (PD) and three of eight with stable disease (SD) had ctDNA levels of several percent during treatment. The remaining patients with SD had either no change or a decrease. Three of four patients with partial response had a decrease of ctDNA to undetectable levels during treatment, the fourth responder had no detectable levels at any timepoint. Moreover, low baseline ctDNA levels predicted long OS (Fig. 1g, Supplementary Data 3), but not PFS (Supplementary Fig. 1c). These levels were also nominally, but not significantly, lower in patients with PR as compared with progressive disease (PD) (Supplementary Fig. 1d). ctDNA levels correlated with LDH (Supplementary Fig. 1e), and low levels of LDH also predicted long OS (Fig. 1h), but not PFS

(Supplementary Fig. 1f). Although, LDH levels were not significantly different between RECIST-response groups (Supplementary Fig. 1g).

**Safety and quality of life**. At baseline, 24 patients had Eastern Collaborative Oncology Group (ECOG) performance status of zero and five had an ECOG of one. About 18 weeks after treatment initiation, 12 patients were at ECOG zero, five at ECOG one, and 12 were missing. Adverse events (AEs), regardless of assessed causality, were reported in 28 patients (97%). Nineteen patients (66%) had grade ≥3 AEs (summarized in Table 2), the most common being increased blood alkaline phosphatase levels followed by neutropenia, increased aspartate/alanine aminotransferases, and rash. Twenty-five patients (86%) experienced an immune-related adverse event (irAE), and eight patients (28%) had an irAE of grade ≥3: Four events of hepatitis, two events of skin toxicity, and one event each of colitis and stomatitis was observed; all grade 3. One event of grade 4 hypophysitis was also recorded. Thirteen patients (45%) received immune-modulating drugs for the management of irAE (corticosteroid monotherapy in twelve patients, and one patient received

**Table 2 Adverse events (AE) of grade ≥3 (n = 29 patients).**

| System Organ Class, Preferred Term | N (%) |
|---|---|
| Patients with any AE of grade ≥ 3 | 19 (66) |
| *Blood and lymphatic system disorders* | |
| Neutropenia | 3 (10) |
| Lymphopenia | 1 (3) |
| *Investigations* | |
| Blood alkaline phosphatase increased | 4 (14) |
| Aspartate aminotransferase increased | 3 (10) |
| Alanine aminotransferase increased | 2 (7) |
| Cortisol decreased | 1 (3) |
| *Skin and subcutaneous tissue disorders* | |
| Rash | 2 (7) |
| *Gastrointestinal disorders* | |
| Colitis | 1 (3) |
| Nausea | 1 (3) |
| Stomatitis | 1 (3) |
| *Metabolism and nutrition disorders* | |
| Hyperglycemia | 1 (3) |
| Hypokalemia | |
| Hyponatremia | 1 (3) |
| *Endocrine disorders* | 1 (3) |
| Hypophysitis | 1 (3) |
| *Respiratory, thoracic and mediastinal disorders* | |
| Pulmonary embolism | 1 (3) |
| *Musculoskeletal and connective tissue disorders* | |
| Back pain | 1 (3) |
| Lupus-like syndrome | 1 (3) |
| *Hepatobiliary disorders* | |
| Jaundice | 1 (3) |
| *Nervous system disorders* | |
| Insomnia | 1 (3) |
| *General disorders and administration site conditions* | |
| Chest pain | 1 (3) |

addition of mycophenolate). Dose interruption and reduction of entinostat was required in nine patients (31%), due to neutropenia (with or without thrombocytopenia) in five patients, nausea in three patients and rash in one patient. Three patients (10%) had an AE leading to treatment discontinuation: one patient each with grade 2 pneumonitis, grade 3 hepatitis, and grade 4 hypophysitis. There were no treatment-related deaths. Quality-of-life assessments did not show any statistically different changes in The Functional Assessment of Cancer Therapy—General (FACT-G) score or the patient's self-rated health status (Supplementary Fig. 2a, b).

**PD-L1 expression and TIL analyses**. Twenty-three patients (79%) had adequate formalin-fixed, paraffin-embedded tissue available for PD-L1 and TIL evaluation (Supplementary Data 4). Three of the 23 evaluable samples could not be evaluated for a PD-L1 interface pattern. Tumor cell PD-L1 expression was identified in only one patient (4.3%), but nine patients (39%) had a PD-L1 tumor-modified percent score (MPS) greater than zero. The patient with PD-L1-positive tumor cells had progressive disease as best overall response. There was no significant association between survival and PD-L1 score (Supplementary Fig. 1h, i) nor any correlation between clinical benefit at week 18 ($p = 0.36$) and a positive MPS or PD-L1 positivity at the stromal interface ($p > 0.99$). All 23 evaluable patients had TILs within tumor nests: six had a score of 1, seven had a score of 2, and 10 had a score of 3. There was no significant correlation between TIL scores and survival (Supplementary Fig. 1j, k) or a high TIL score (3) and clinical benefit ($p = 0.65$).

**Genetic analyses**. DNA and RNA were extracted from pretreatment formalin-fixed biopsy specimens. Exome- and RNA sequencing was performed on samples passing quality control criteria (22 DNA samples and 20 RNA samples). As expected, driver mutations were detected in *GNAQ* and *GNA11* (Fig. 2a) and the tumor suppressor gene *BAP1*, which coincided with chr 3 monosomy (Fig. 2b). There were no statistically significant differences in survival between patients with different HLA allelic diversity (Supplementary Fig. 3a, b) or between patients with *GNAQ* or *GNA11* tumor mutations. However, similar to in primary UM, patients with *GNAQ*-mutated tumors had a nominally better survival (Fig. 2c, d). Of the four patients responding to entinostat and pembrolizumab, three had wild-type *BAP1*, resulting in a significant association between partial response and *BAP1* wild-type status (Fig. 2e). Patients with wild-type *BAP1* also had nominally longer OS and PFS than those with mutations (Supplementary Fig. 3c, d). Tumor burden might be somewhat higher in *BAP1*-mutated tumors, as suggested by ctDNA and LDH levels, although differences in these metrics were not significant (Supplementary Fig. 3e, f). The fourth patient with a partial response had a *GNA11*-mutant iris melanoma with a mutational signature indicative of UV damage, which resulted in an outlier tumor mutational burden (Supplementary Figs. 3g and 2f). The anterior position of the iris is likely to be susceptible to UV damage, as we and others recently demonstrated[13,42]. Interestingly, just like cutaneous melanoma (where the majority of tumors also have a UV signature and respond to PD-1 inhibition), the patient responded to treatment in the PEMDAC trial (Figs. 1a–c and 2g, all other responders shown in Supplementary Fig. 4). Patients with wild-type *BAP1* or UV signature UMs survived longer than patients with tumors that had mutant *BAP1* and no UV signature (Fig. 2h, i).

**Blood analyses**. Immune cell compositions were analyzed before and after treatment by flow cytometry in 24 patients to investigate therapy-related changes in immune cell composition between patients with shorter (equal to or below median) and longer (above median) OS. PBMCs collected before and after one cycle of treatment were analyzed by flow cytometry. The total number of T cells increased nominally after treatment in patients with the longer survival but not in patients with the shorter survival (Fig. 3a, b). An increase in monocytes was observed in both groups, among the longer-survival group, this was accompanied by a decrease in neutrophils following treatment (Fig. 3a, b). Irrespective of the outcome, there was a significant difference in activated CD8$^+$ T-cell frequency following treatment (Fig. 3c). There were no changes in CD4$^+$ T cells expressing CXCR3, CCR4, CCR6, and CXCR5 and CD127$^-$/CD25$^+$ serving as surrogate markers of Th1, Th2, Th17, Tfh and T-regulatory subsets, respectively, between groups or following treatment (Supplementary Figs. 5 and 6a, b). Therefore, there were changes in immune cell population proportions in patients treated with combined entinostat and pembrolizumab.

Blood from three patients with progressive disease (1–001, 1–005, and 1–006) was drawn before and after one treatment cycle and the immune cells analyzed by single-cell sequencing of the T-cell receptor (TCR) and mRNA. Cells were clustered into cell types by gene expression similar to the flow cytometry, but there were no major differences in the blood compositions of these samples following treatment (Fig. 3d, Supplementary Fig. 5c). TCR clonotyping did not reveal expansion of any specific T-cell clones. Just like in UM cells, T cells, NK cells, and to some extent monocytes from entinostat/pembrolizumab-treated patients had elevated HLA gene expression (Fig. 3e). Genes encoding T cell activation/exhaustion markers, such as

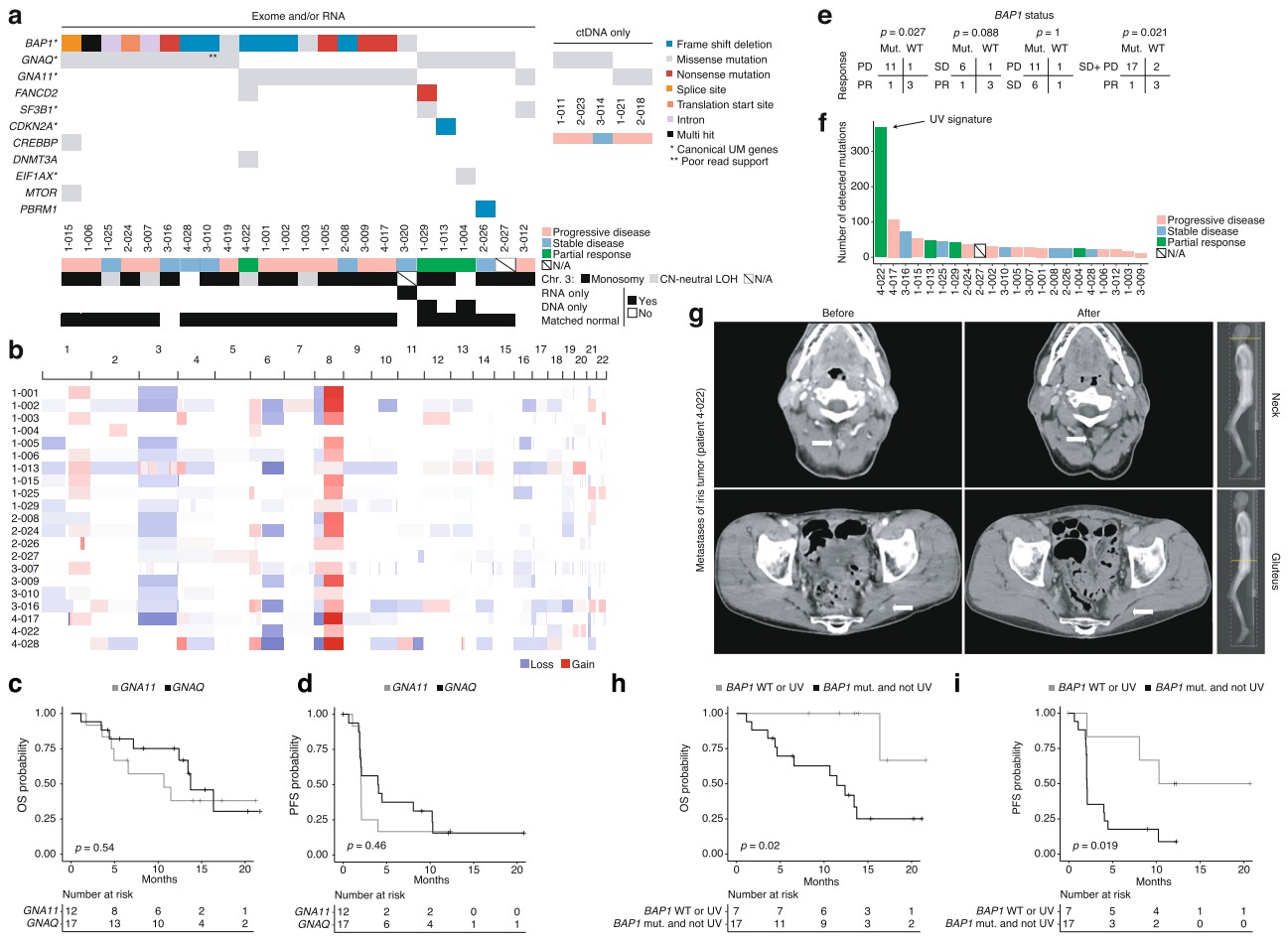

**Fig. 2 Genetic analyses of pretreatment biopsies from patients recruited to the PEMDAC trial. a** Mutations in genes that are either recurrently altered in UM or listed among COSMIC Cancer Gene Census driver genes. Responses in the trial are indicated. **b** Copy number profiles of each tumor inferred from exome sequencing data of tumors and matched normal tissue. Differences in color intensity depend on copy number amplitude and tumor purity. **c**, **d** OS and PFS analyses comparing patients with *GNAQ*- or *GNA11*-mutated UM (*n* = 21). **e** Fisher's exact test of *BAP1* mutational status versus responses in the PEMDAC trial. PD: progressive disease; SD: stable disease; PR: partial response. **f** Number of detected mutations from exome-sequencing data. **g** CT scans of patient 4–022 at baseline and at best response (22 months post therapy). Arrows show PET-positive lesions that disappeared from the dorsal neck region and left gluteus. **h**, **i** Comparison of OS and PFS to assess if UM patients with a wild-type *BAP1* status or a UV-damaged genome survive longer than the other patients (*n* = 24). In (**c**, **d**) and (**h**, **i**), *p*-values for survival associations were calculated using log-rank tests. No adjustments for multiple comparisons were made. All statistical tests were two-sided. Source data are provided as a Source Data file.

PD-1, CTLA-4, TIM-3, and TIGIT were also induced. Intriguingly, other upregulated genes included some of those induced by entinostat in UM cells[31] such as *FOS*, *JUN*, and *NR4A2*. These changes do not appear to be primarily caused by changes in T-cell clones, since they were also present within identical clonotypes before and after treatment in a given patient (Supplementary Fig. 5d). Therefore, these are likely to be entinostat-induced changes. Collectively, blood analyses showed that entinostat and pembrolizumab induce changes in lymphocytes, monocytes, and neutrophils in patients with longer survival and that target engagement of the HDAC inhibitor can be measured as specific gene expression changes in T cells in blood samples.

## Discussion

There are currently no FDA- or EMA-approved systemic therapies for patients with metastatic UM, who have a dismal prognosis[3]. PD-1 inhibition has transformed the management and prognosis of patients with metastatic cutaneous melanoma. Although most existing data on immunotherapy in metastatic UM have been gathered from retrospective analyses of off-label use, producing variable and possibly biased results, their efficacy

has been disappointing. Despite a lack of prospective data, we hypothesized that PD-1 inhibitors are insufficiently effective as monotherapy in UM and that combination therapy would improve outcomes. Previous studies from ourselves and others have shown that HDAC inhibitors modulate immune gene expression in cancer, including in HLA genes[30,31,43]. This is likely to be via inhibition of a class 1 HDAC given the selectivity profile of entinostat. We chose to use entinostat (over vorinostat) because of its longer plasma half-life and favorable pharmacodynamic profile. We have previously shown in mouse melanoma in vivo and human UM in vitro, that entinostat monotherapy induces PD-L1 in cancer cells[31]. This may counteract any beneficial immunotherapeutic effects of HDAC inhibition. Indeed, entinostat-treated B16–F10 melanoma cells grew faster in vivo, an effect reversed by CRISPR-mediated *Cd274* (*Pdl1*) knockout using CRISPR or anti-PD1 treatment[31]. This provided a strong rationale to test HDAC and PD-1 inhibition in the clinic to leverage the positive immune-stimulatory effects of both drugs.

Previous immunotherapy—including checkpoint inhibitors, adoptive cell therapy or bispecific agents such as Tebentafusp - was the main exclusion criterion in the PEMDAC trial. During

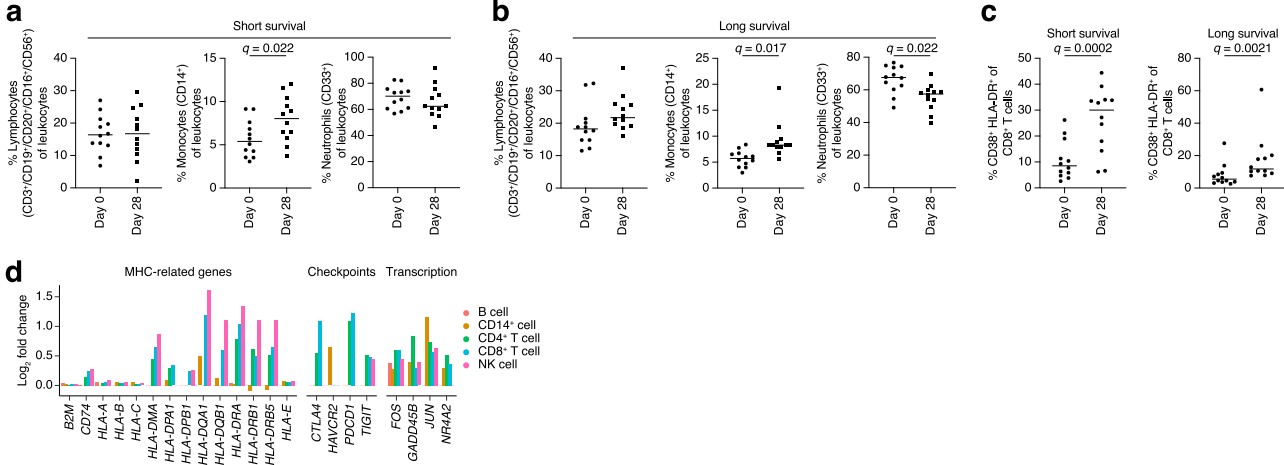

**Fig. 3 Immune profiles of pre- and post-treatment (one cycle) blood samples. a–c** Flow cytometry analyses of changes in circulating cell populations in shorter and longer survivors following treatment with entinostat and pembrolizumab. Gating strategies can be found in Supplementary Fig. 5. (**a**) Graphs showing the frequency of CD3 + lymphocytes (left), CD14 + monocytes (middle), and CD33 + neutrophils (right) among patients with shorter (n = 12) and (**b**) longer (n = 12) survival prior to (circles) or after (squares) one cycle of treatment. (**c**) Analysis of CD8 + cytotoxic T cells in patients with shorter and longer survival. Statistical analysis was performed using multiple paired t-tests correct for multiple comparisons using a two-stage step-up (Benjamini, Krieger, and Yekutieli) with test between all groups in (**a–b**), where * indicates adjusted p-values < 0.05. **d** White blood cells from three patients were analyzed by 10x Genomics TCR and gene expression analysis. Statistically altered genes in different cell types after treatment in the categories antigen presentation, immune-checkpoint receptors, and transcriptional regulation. n = 3 patient samples were compared pre- and post treatment. Genes shown were significant at q < 0.05. p-values were calculated with Wilcoxon rank-sum tests and adjusted for multiple comparisons with the Benjamini–Hochberg method. All statistical tests were two-sided. Source data are provided as a Source Data file.

enrollment over 10 months in Sweden, the national guidelines did not recommend checkpoint inhibitors for uveal melanoma, leaving the majority of metastatic UM patients in the country eligible for participation in the trial. The patient cohort was clinically heterogeneous and included both previously treated and treatment-naive patients. Despite this, combined entinostat and pembrolizumab produced durable responses with manageable toxicities in UM patients. The primary endpoint was met with an ORR of 14%. This compares favorably with reports of anti-PD-1 monotherapy, with the largest case series to date reporting an ORR of ~5%[16,17]. The median OS of 13.4 months was longer than the ten-month benchmark survival determined from historical data, although possible differences in patient characteristics demand caution in making direct comparisons[9]. While the observed median PFS of only 2.1 months appears short, median PFS is highly sensitive to the chosen evaluation schedule in treatments where the most common best overall response is PD, and does not reflect possible long-term benefit for a minority of responding patients in immune oncology trials.

Although combined entinostat and pembrolizumab treatment was associated with a high incidence (66%) of grade ≥3 AEs, the most frequent AEs were abnormal laboratory investigations that were not necessarily clinically relevant and were in some cases related to disease progression. Others were expected hematological toxicities from entinostat easily manageable with dose interruptions or reduction. The incidence of irAEs (86%) in our study appears higher than that predicted from experience with single-agent PD-1 inhibitors in cutaneous melanoma, which is typically around 50%[44]. The occurrence of irAEs has been shown to correlate with anti-PD-1 treatment effect in several diseases[45,46], and one may speculate that the observed incidence of irAEs found here might be indicative of immune activation and possibly increased efficacy through the addition of entinostat. Reassuringly, only 34% of patients experienced a severe irAE (grade ≥ 3), making the treatment manageable and safe in centers with experience in handling of irAEs. Importantly, there were no treatment-related deaths.

Data from multicenter phase II trials of ipilimumab and nivolumab (ipi–nivo) showed an ORR of 12%[47,19] and 18%[18] comparable to that reported in a retrospective multicenter study[48]. All three studies also reported an OS and PFS with ipi-nivo that compared favorably to historical data[9], partly explaining the widespread use of this regimen in routine clinical practice. With a very high rate (50–60%) of severe (grade ≥ 3) treatment-related AEs[49], the ipi–nivo regimen is not suitable for all patients and treatment-related deaths have been described[47]. Nevertheless, it is highly likely that patients with metastatic UM in some countries will receive ipi–nivo and/or the bispecific agent Tebentafusp, albeit this agent only works for patients with an HLA-A2 haplotype[15]. It is not known whether or not combined PD-1 and HDAC inhibition would be useful on progression following these agents, but phase I/II trials have shown encouraging results for entinostat and pembrolizumab in patients with PD-1 inhibitor-refractory cutaneous melanoma or lung cancer[36,37]. Whether a potential mechanism of response in UM would be by targeting a different subset of patients or actual reactivation of immunity will be of future interest to study.

Moving forward, for the clinical development of combined checkpoint immunotherapy and epigenetic therapy, it was of upmost importance to identify tumor characteristics that correlated with response and match these to those of other immunotherapy treatments. No clear biomarkers of response have been disclosed from the ipi–nivo studies, but we found several features of the tumors in this study that could be of interest to also investigate as putative biomarkers in the context of ipi–nivo or Tebentafusp trials. First, low levels of baseline ctDNA correlated with long overall survival but not progression-free survival. This result confirms very recent data in patients with cutaneous melanoma treated with checkpoint inhibitors[50,51]. The reason why levels of ctDNA predict survival is not clear but may well be associated with tumor burden as low LDH levels also correlated with longer survival in this trial. In a previous small study using pembrolizumab in UM, a response or disease stabilization was seen in patients with low tumor volume[52]. Mechanistically, a high

tumor burden and release of vesicles (carrying ctDNA) could suppress antitumor immunity, thereby impacting therapeutic response of immunotherapy[53].

Second, we show for the first time that a patient with metastatic uveal melanoma originated from the iris, a very rare condition, responds to immunotherapy. The iris melanoma in the patient had an outlier TMB due to a UV-damaged genome. This finding is consistent with two recent studies demonstrating that iris melanomas have UV-damaged genomes[13,42]. Similarly, two other studies show that germline mutations in the DNA repair protein MBD4 can cause a higher TMB and response to anti-PD1 therapy in UM[54,55]. We did not detect MBD4 mutations in any of the tumors in the present study, and only the iris melanoma had high TMB. Given these case descriptions and that these patients are too rare for separate clinical trials, checkpoint inhibitor therapy is worth considering in patients with UM with high TMB. Indeed, pembrolizumab is FDA-approved for tumors with a high TMB, but a companion diagnostic for MBD4 or UV-related high TMB has yet to be validated.

Third, three patients with BAP1 wild-type tumors responded to combination therapy. BAP1 is a multifaceted protein which, when mutated, is associated with poor survival in analyses of primary UM[12,56]. However, this is the first report of an association between BAP1 mutation status and treatment outcomes in metastatic UM. While BAP1 has many functions[57], it is tempting to speculate that BAP1 regulates UM immunogenicity and that its mutational status is both prognostic and predictive. While recent immune profiling studies of primary and metastatic UM would favor BAP1 regulating an immunosuppressive microenvironment[58], it cannot be excluded that responses to HDAC inhibition differ between tumors with different BAP1 status. However, previous studies of UM and mesothelioma cells have shown that BAP1 inactivation increases sensitivity to HDAC inhibitors and reprograms their stemness[28,59–61]. On the other hand, trials of HDAC inhibitor monotherapy are either not published (NCT01587352) or have not shown efficacy in few patients recruited with UM[62]. Therefore, the clinical relevance of enhanced sensitivity of BAP1-deficient cells to HDAC inhibition alone in vitro may be low. Rather, it is plausible, given that BAP1 wild-type tumors appear to be more sensitive to combined HDAC and PD-1 inhibition in this trial, that the immune modulatory functions of HDAC inhibitors rather than their direct tumor cell-killing effect may be dominant.

To summarize, here we show for the first time that combined epigenetic and immunotherapy can cause tumor regression in a small subset of patients with metastatic UM. Low tumor burden (ctDNA and LDH), presence of the tumor suppressor BAP1, and an outlier tumor mutational burden in an iris melanoma correlated with response and/or survival, possibly highlighting how outcomes for patients with UM metastases are highly dependent on the intrinsic tumor genetics and the tumor microenvironment. Combined epigenetic and immunotherapy would be interesting to evaluate in a randomized trial compared with immunotherapy alone. This would also address whether BAP1 mutational status is predictive of immunotherapy response or only in combination with epigenetic therapy. If entinostat would impact combined nivolumab and ipilimumab treatment, tebentafusp treatment or locoregional therapy of patients with UM is not known neither. These studies are warranted since uveal melanoma continues to be a difficult-to-cure disease.

## Methods

**Clinical trial**. The study protocol and all amendments were approved by the regional ethical review board in Gothenburg (#692–16) and the Swedish Medical Products Agency (EudraCT registration number: 2016–002114-50). Signed and dated informed consent was obtained from each patient in accordance with the principles of ICH-GCP and the latest version of the Declaration of Helsinki.

ClinicalTrials.gov registration number: NCT02697630 (registered March 3, 2016). The study protocol is available in an open-access publication[63].

**Patients**. Eligible patients were aged ≥18 years and had histologically or cytologically confirmed metastatic UM; measurable disease by computed tomography (CT) or magnetic resonance imaging (MRI) as per RECIST 1.1 criteria; ECOG performance status 0–1; and could have received any number of prior therapies (including none), with the exception of anticancer immunotherapy. Key exclusion criteria were active brain metastases, active autoimmune disease, immunodeficiency or treatment with systemic corticosteroids, previous treatment with anticancer immunotherapy, use of other investigational drugs within four weeks before study drug administration, and life expectancy of less than three months. The first patient was enrolled on 21 February 2018 and the last patient was enrolled on 21 December 2018.

**Study design**. PEMDAC is a phase-II, single-arm, multicenter study. The study was investigator initiated and carried out at the four major Swedish university hospitals with support of the Swedish Melanoma Study Group (SMSG). The planned sample size was 29 patients allocated using Simon's optimal two-stage design. At least one confirmed response among the first ten patients was required to enroll the additional 19 patients.

Patients were treated with pembrolizumab 200 mg intravenously every third week in combination with entinostat 5 mg orally once weekly. Treatment continued until documented disease progression, intolerable side effects, patient's withdrawal of consent, or decision of the investigating physician to end treatment, or to a maximum of two years. Treatment beyond progression was allowed if the patient was clinically stable according to the criteria specified in the study protocol[63].

The objective was to determine if combined treatment with entinostat and pembrolizumab could have clinical efficacy in metastatic UM patients. The primary endpoint was objective response rate (ORR) according to RECIST v1.1 criteria[64]. The secondary endpoints included clinical benefit rate (CBR) at week 18, overall survival (OS), progression-free survival (PFS), and incidence and severity of adverse events (AEs). Exploratory endpoints included response by immune-related RECIST (irRECIST) criteria[65] and extensive biomarker analyses.

Radiological assessments were performed every nine weeks. Adverse events (AEs) were registered and graded according to CTCAE v4.03. Blood and tissue for biomarker analyses were collected throughout the study.

**Statistical analysis**. Efficacy and safety analyses included all patients who received one dose of study treatment. The sample size and power estimates were based on the primary endpoint ORR alone. Power was required to be 80% at a significance of 5%. We assumed that an ORR of 5% was not a clinically relevant treatment effect, whereas 20% was sufficient to consider the treatment useful. Patients were enrolled in two batches, the first consisting of ten patients and the second group of 19, the optimal allocation according to Simon's optimal two-stage design (significance level =5%, one-sided)[66]. The study was considered positive if at least four patients out of 29 had a confirmed objective response. Proportional outcome measures are reported using 95% confidence intervals (CI). Since the sample size was small, an exact method was used. As applicable, tests were conducted versus zero or the highest noneffient value. Times to various events were analyzed using nonparametric methods. Time was summarized using medians through the Kaplan–Meier method together with 95% CIs. Fisher's exact test was used to assess associations between immunohistochemical analyses (PD-L1 and TILs) and clinical benefit.

**Immunohistochemical analyses**. Tumor PD-L1 testing was performed at a central laboratory (QualTek Molecular Laboratories, Newtown, Pennsylvania). Formalin-fixed, paraffin-embedded (FFPE) baseline tumor samples were evaluated using the 22C3 antibody (Merck & Co., Inc., Kenilworth, NJ). Interpretation was performed using a modified proportion score (MPS) indicating the proportion of PD-L1-expressing mononuclear inflammatory cells (MICs) plus PD-L1-positive tumor cells within tumor nests, as well as by the presence of a distinctive PD-L1 staining pattern at the tumor–stroma interface. Morphological evaluation of hematoxylin and eosin (H&E)-stained sections was used to identify TILs (graded from 0 to 3) and MICs, assess sample quality, confirm the diagnosis, and establish tumor burden.

*DNA and RNA sequencing*. DNA and RNA were prepared from FFPE sections from patients in the PEMDAC trial using the Tissue FFPE DNA/RNA kit (Qiagen). Nontumor DNA was extracted from PBMCs using a. Exome and RNA sequencing was performed at the Genome Medicine Center at Sahlgrenska University Hospital, Gothenburg, Sweden.

*Cell-free circulating DNA sequencing*. Whole blood was collected and plasma was kept at −80 °C. Before extraction of circulating tumor DNA, plasma was centrifuged at $16,000 \times g$ at 4 °C for 20 min with an Eppendorf 5804 R centrifuge to remove cellular debris. Subsequently, cfDNA was extracted using the QIA-Symphony DSP Circulating DNA kit and eluted in 60 μL of AVE buffer (Qiagen, Hilden, Germany), quantified using the Qubit dsDNA HS kit (Thermo Fisher

Scientific), and cfDNA was stored at −20 °C. The cfDNA was concentrated using Vivacon 500 30,000 molecular weight cut-off reverse-spin columns (Sartorius, Göttingen, Germany) before SiMSen-Seq libraries were generated as previously described[67]. Briefly, in a first PCR, assays specific for the mutated regions of the UM oncogenes *GNAQ*, *GNA11*, *CYSLTR2*, and *PLCB4* were used to incorporate molecular barcodes onto sample molecules. In a second PCR, sequencing adapters were introduced and final libraries subsequently purified using the Agencourt Ampure XP system (Beckman Coulter, Brea, CA). Library quality and size distribution were assessed using the Fragment Analyzer HS NGS kit (Agilent Technologies, Santa Clara, CA). Libraries were then pooled and quantified by qPCR using a modified version of the NEBNext Library Quant Kit for Illumina (New England BioLabs, Ipswich, MA). Clustering was performed at 1.8 pM on a MiniSeq instrument in 1 × 150 bp mode supplemented with 10% Phix Control v3 using a 150 bp High Output Reagent Cartridge (all Illumina, San Diego, CA). Raw FASTQ files were analyzed using a modified version of debarcer[67] on an Ubuntu 18.04.3 LTS cluster and aligned to hg38 using bwa[68]. Briefly, valid reads within each amplicon were identified as those containing a barcode sequence in the correct position relative to the hairpin structure. Reads were then grouped into families by amplicon and barcode. For reads within each family, a consensus sequence was determined for each base. Nonreference sequences were reported in consensus sequences if they composed 100% of the reads in families with 3–20 reads or at least 90% of reads in families with >20 reads.

*Preprocessing of DNA-seq data.* Exome sequencing reads were aligned to the 1000 Genomes version of the hg19 human reference genome (v. 37) with bwa (v. 0.7.17)[68] using the arguments "mem -t 10 -M -R". Alignments corresponding to multiple sequencing runs of the same sample were merged using the samtools "merge" command (v.1.9)[69]. Duplicate reads were marked with MarkDuplicates (GATK v. 4.1.3.0)[70] using default parameters. Base-quality score recalibration was performed with BaseRecalibrator and ApplyBQSR (GATK) in two passes using the same reference genome, as well as lists of known polymorphisms from the GATK resource bundle (files "dbsnp_138.b37.vcf", "1000G_phase1.indels.b37.vcf" and "Mills_and_1000G_gold_standard.indels.b37.vcf").

*Preprocessing of RNA-seq data.* RNA-seq reads were aligned to the 1000 Genomes[71] version of the hg19 human reference genome (v.37) with STAR (v.2.7.1a)[72]. Arguments used were "–twopassMode Basic–outFilterType BySJout–sjdbOverhang 100–outSAMmapqUnique 60", and the NCBI GRCh37.75 reference genome annotation was provided as a database of known splice junctions. Read counts per gene were obtained from alignments using htseq-count (HTSeq v. 0.11.2)[73] with the arguments "-r name -q -f bam -s reverse -m intersection-strict" as well as the NCBI GRCh37.75 reference genome annotation.

*Variant calling for DNA-seq data.* Variant calling for exome-sequencing alignments was performed with Mutect 2[74] (GATK v. 4.1.3.0) using the parameters "–genotype-germline-sites true–genotype-pon-sites true–af-of-alleles-not-in-resource 0.0000025–disable-read-filter MateOnSameContigOrNoMappedMateReadFilter". The GnomAD[75] population variant database was provided as a germline resource, together with the same reference genome as above. The analysis was restricted to exome target regions corresponding to Agilent SureSelect Clinical Research Exome v2 or Twist Exome, depending on sequencing batch. In addition, a panel of normals was supplied as input, built from all available normal samples in the study. This panel was built by first running Mutect 2 in tumor-only mode on each normal with the parameter "–disable-read-filter MateOnSameContigOrNoMappedMateReadFilter" and then running CreateSomaticPanelOfNormals (GATK) on the resulting files. Variant-quality labels were assigned using FilterMutectCalls (GATK) using the same reference genome as previously. These variants were then annotated using the script vcf2maf.pl (https://github.com/mskcc/vcf2maf), which relies on VEP[76], using the v. 98 build of the VEP reference database for the GRCh37 genome.

*Variant calling for RNA-seq data.* Duplicate reads were marked with MarkDuplicates (GATK v. 4.1.3.0) using default parameters. Reads spanning splicing events were split using SplitNCigarReads (GATK), with the 1000 genomes version of the hg19 human reference genome (v37) provided using the "-R" parameter. Base-quality score recalibration was performed as described for exome data. Variant calling was performed with HaplotypeCaller (GATK) using the same reference genome and the arguments "–dont-use-soft-clipped-bases = true–standard-min-confidence-threshold-for-calling = 20.0". Variant-quality labels were assigned using VariantFiltration (GATK) using the arguments "–cluster-window-size 35–cluster-size 3–filter-name FS–filter-expression "FS > 30.0"–filter-name QD–filter-expression "QD < 2.0". Variants passing these criteria were annotated as described for exome data.

*Mutational signature analysis.* To determine mutational spectra, all somatic autosomal mutations (including synonymous) not present in any population variant resource and with minor allele read support ≥5 were converted into a 96-trinucleotide mutation-frequency matrix using the function *mut_matrix* from the R package *MutationalPatterns* (v. 3.0.1)[77] with the parameter "ref_genome = 'hg19'". Known mutational signature

trinucleotide frequencies, obtained via COSMIC (http://cancer.sanger.ac.uk/cancergenome/assets/signatures_probabilities.txt; accessed Feb. 10, 2020), were then fitted to the observed mutations using the function *fit_to_signatures*. This algorithm operates by searching for the nonnegative linear combination of the predefined mutational signatures that best explains all mutations in a given sample, which is achieved by solving a nonnegative least-squares optimization problem. This results in estimates of the relative contributions of known mutational signatures in each sample.

*Copy number analysis.* Copy number segmentation was performed with CNVkit (v. 0.9.6)[78]. First, the command *cnvkit.py batch* was used, specifying matching tumor and normal files (where available), exome target regions based on the kit used (Agilent SureSelect Clinical Research Exome V2 or Twist Exome), the 1000 Genomes version of the hg19 human reference genome (v37), and a list of problematic regions to exclude (http://hgdownload.cse.ucsc.edu/goldenpath/hg19/encodeDCC/wgEncodeMapability/wgEncodeDukeMapabilityRegionsExcludable.bed.gz). The resulting output was converted to SEG-formatted files using the commands *cnvkit.py segmetrics* (parameters: "–ci -a 0.05") followed by *cnvkit.py call* (parameters: "–center "median"–purity 1–filter ci") and *cnvkit.py export seg*.

*Single-cell RNA-seq analysis.* Alignment and estimation of gene expression levels were performed with Cell Ranger (v. 3.0.2, 10x Genomics). The specific commands used were *cellranger count* (with the 10x Genomics version of the GRCh38 reference transcriptome; v. 3.0.0) and *cellranger vdj* (with the 10x Genomics GRCh38 VDJ reference dataset; v. 2.0.0).

Clustering and determination of cell types was performed with the *metacell* R package[79]. For this purpose, cells with UMI counts below 800 were ignored and immunoglobulins, mitochondrial, and cellular stress-associated genes were excluded. A gene set was defined to serve as a basis for clustering comprising genes with scaled variances >0.08, total UMI counts >200, and >2 UMI in at least three cells. From this set, genes with correlations >0.1 to a set of cell cycle- and gender-associated genes (*MKI67*, *HIST1H1D*, *PCNA*, *SMC4*, *MCM3*, *TOP2A*, *XIST*, *TSIX*, and *ZFY*) were further excluded. A K-nn graph using the remaining genes was then constructed (function *mcell_add_cgraph_from_mat_bknn*, parameter: "K = 200") and metacells computed (functions *mcell_coclust_from_graph_resamp*, parameters: "min_mc_size = 20, p_resamp = 0.75, n_resamp = 500"; *mcell_mc_from_coclust_balanced*, parameters: "K = 30, min_mc_size = 30, alpha = 2"). A layout for the visualization of the resulting metacells/clusters was made with the *mcell_mc2d_force_knn* function. Subsequently, fold-enrichment scores of selected marker genes (*CD8A*, *CD8B*, *CD4*, *CD3G*, *TIGIT*, *FOXP3*, *CTLA4*, *TRGV9*, *TRDV2*, *TRDV3*, *CD14*, *CD19*, *NCAM1*, *NCR1*, *HBA1*, and *HBB*) were investigated across metacells to identify cell types.

After identifying cell types, the remaining analyses were performed using the *Seurat* R package (v. 4.0.3):[80] data were imported and then normalized using the *NormalizeData* function with default settings. Differential expression between samples was assessed using Wilcoxon rank-sum tests. Cells with more than one TCR alpha to beta chain and other cells predicted to be duplicates were excluded from statistical tests with the R package *DoubletFinder* (v. 2.0.3, parameters: PCs = 1:15, pN = 0.25). Genes expressed in at least 50% of cells in either condition were tested. Adjusted *p*-values (Benjamini–Hochberg correction) <0.05 were considered statistically significant.

*Survival analysis.* Kaplan–Meier curves were produced using the functions *ggsurvplot* and *surv_fit* from the *survminer* R package (v. 0.4.9)[81], and *p*-values for survival associations were calculated using log-rank tests with the *coxph* function from the *survival* package (v. 3.2–11), with the parameter "ties = 'exact'".

**Reporting summary**. Further information on research design is available in the Nature Research Reporting Summary linked to this article.

## Data availability
Source data are provided with this paper. Sequencing data that support the findings of this study have been deposited in European Genome-phenome Archive (EGA) with the accession code EGAS00001005478, under restrictions of controlled access. Figures with associated sequencing raw data are Fig. 2a–f, h–i, Fig. 3d, Supplementary Figs. 3 and 6 c–e. Online resources and databases used in this study include COSMIC (http://cancer.sanger.ac.uk/cancergenome/assets/signatures_probabilities.txt), GnomAD (https://gnomad.broadinstitute.org/), Encode (http://hgdownload.cse.ucsc.edu/goldenpath/hg19/encodeDCC/wgEncodeMapability/wgEncodeDukeMapabilityRegionsExcludable.bed.gz), and dbSNP (https://www.ncbi.nlm.nih.gov/snp/). Source data are provided with this paper.

## Code availability
Code is available at bitbucket.org/jowkar/pemdac_code.

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

## Acknowledgements

The PEMDAC clinical study was sponsored by Sahlgrenska University Hospital and the Västra Götaland Region. The study was carried out within the Merck Investigator Initiated Programme (MISP) and received financial support from Merck Sharp & Dohme Corp and Syndax Pharmaceuticals to L.N. Additional generous grant support came from Cancerfonden (to J.A.N. and A.S.), Familjen Erling Persson (to J.A.N.), Knut and Alice Wallenberg Foundation (to J.A.N. and A.S.), Vetenskapsrådet (to J.A.N. and A.S.), Sjöbergstiftelsen (to J.A.N.), BioCARE Strategic grants (to J.A.N.), Lion's Cancerfond Väst (to J.A.N., A.S., L.N. and S.F.), Västra Götaland Regionen ALF grant (to J.A.N., A.S. and L.N.), Assar Gabrielsson Fond (to V.S. and S.F.), Vinnova (to A.S.), Johan Jansson Stiftelse (to S.F.), and Gustaf V Jubileumsklinikens forskningsfond (to L.N.). We thank Carina Karlsson and Helena Kristiansson for technical support, Eren Svensson for study monitoring, Anette Eriksson for study coordination and documentation, Martin Nilsson for radiological support, and Maria Persson, Statistikkonsulterna for statistical support.

## Author contributions

L.N., H.J. and J.A.N. designed the study. J.A.N. and L.M.N. conceived the study. H.J. and J.A.N. wrote the main texts of the paper. J.K., S.A. and S.F. wrote experimental descriptions. All authors contributed to editing. J.K. performed all bioinformatics analyses, all exploratory statistical analyses and generated figures. S.A., V.S., S.F. and H.J. managed biobanking and/or conducted blood analyses. B.A, L.M.N. and A.S. supervised biobanking, genetic and blood analyses. L.N. was the PI of the clinical trial. L.N., H.J., C.A-E, A.C., H.H., M.L., I.L., R.O.B., U.S. and G.U. contributed to recruitment, management and follow-up of patients in the clinical trial.

## Funding

## Competing interests

Lars Ny received a research grant from MSD and Syndax Pharmaceuticals to support some aspects of the PEMDAC study. Anders Ståhlberg is the coinventor of SiMSen-Seq (ctDNA measurement), is a board member of, and has stock ownership in SiMSen Diagnostics. The remaining authors declare no conflict of interest.
