## [Peer Review File · Nature Communications]

Reviewers' Comments:

Reviewer #1:

Remarks to the Author:

Ny, Jespersen et al present a very interesting Phase 2 trial of combined HDAC and PD-1 inhibition in uveal melanoma. There is a clear unmet need and the trial was well run. They have presented some interesting correlates suggesting that, among other things, non-BAP1 mutant UM may be responsive to this therapeutic approach. I have some suggestions that may improve the manuscript, below. Sincerely, Alex Shoushtari.

- 1) Safety – please elaborate on dose reductions/omissions and steroid or secondary immune suppression usage to give a better broader idea of safety and tolerability of the combo.
- 2) Gene expression analysis appears well done, and 12 genes significant – but can you include a sentence on themes within those genes? This would introduce and better prepare the reader to understand the discussion focusing on these genes. Also, if improved antigen presentation was one of the purported mechanisms of synergy between HDAC inhibition and PD-1 blockade, do we have a hypothesis as to why no antigen presenting machinery or Class 1 HLA genes are in this subset (to the best of my ability to scroll through the supp table)?

Minor

- 1) P. 6 lines 173-74 – “responded to” what? PD-1 blockade?
- 2) When referring to dichotomous OS for PBMC analysis, “longest” and “shortest” imply a subset of extreme responders, whereas you are actually just comparing “longer” and “shorter” OS. Subtle but important distinction to better reflect the comparison.

Reviewer #2:

Remarks to the Author:

In this manuscript, Ny and Colleagues are describing the PEMDAC phase 2 trial associating pembrolizumab and entinostat in patients with metastatic uveal melanoma (mUM). They provide their initial clinical findings, as well as exploratory biologic analyses. This manuscript contains a number of interesting data, well described. However the level of evidence is most often low, and it is difficult to draw any definitive conclusion from this report.

1) Clinical trail

The authors report the outcomes of 28 pts (one excluded from the study) with mUM treated in first line or not with Pembro/Entinostat in a single arm study. ORR was 14%, reaching primary endpoint, by with only PR (no CR). This ORR compares favorably with historical series of Pembro alone (ORR ~5%). However, the incidence and intensity of adverse events (AEs) was very high, especially when compared to historical series of Pembro, leading to the treatment discontinuation of 3 pts and 19 pts (66%) with \geq grade 3AEs (grade 3, 4 and 5 AEs corresponding to AEs ranging from “severe”, to “life-threatening” and “death related to AE”, respectively; the information about the precise grade should be given in this work).

A paradox of mUM is the very long period of no or very mild clinical symptoms of the metastatic disease, in contrast with the very short survival. One may be reluctant to propose such drug combination for such limited benefice. Of note, classically in 3+3 phase I studies, the RP2D (recommended dose for phase II studies) is the dose producing ~20% of dose-limiting toxicity. Nevertheless, PR of 3 out of 4 BAP1wt patients is intriguing, counterintuitive (BAP1mut UM were predicted to be more “immune-responsive”) and quite interesting. However, this observation is a posteriori and needs to be confirm on additional BAP1wt pts, a difficult trial to set-up. One case is easily explained by the high mutation burden associated with the iris origin of the UM. In this case, we can suppose that Pembro would have been sufficient to obtain a tumor response. I would be interested in seeing images of tumors responding to this combo in other patients that the iris UM one.

The observation that LDHlow ctDNAlow predicts longer OS is neither new, nor counterintuitive (less tumor mass, less aggressive disease, better survival).

Minor points:

L132: Levels of circulating tumor DNA corroborated the response findings. Actually, the ctDNA

evolution looks more complex than that, and more analyses and comments should be implemented.

Supp Fig 1. Please develop abbreviations (PD PR SD BOR,...) in the legends. The Journal is for general audience, not sole clinicians.

Table 2. "Adverse events (AE) of grade ≥ 3 in all patients (n=29)." n is ambiguous. Please replace by "Adverse events (AE) of grade ≥ 3 (n=29) in all patients" or similar.

2) Exploratory analyses

These explorations are quite large, but not reached any level of novelty or of significance:

No correlation with outcome was found with PDL1 and TIL analyses

Genetic analyses identified an iris UM with high TMB, which was already reported. The clinical response of this type of patient (TMB is the same order than cutaneous melanoma) is not unusual under Pembro and the role of Entinostat in this response is not clear. The major finding is the enrichment of BAP1wt in responders. However, this provocative finding was done a posteriori finding and needs to be validated in an independent cohort for having significance.

Similarly, the predicted gene expression signature of 12 genes has just no scientific value if not validated in an independent cohort, and its report here is misleading, despite potential information it could contain.

Blood analyses

This exploratory analysis has very little significance. The authors analyzed by multicolor FACS samples before and after treatment, measuring dozens of cell populations. The description of selected blood populations should take into account this hidden multiple testing, and I predict the marginal differences observed between responders and non-responders will not hold appropriate and mandatory statistical corrections.

The authors followed this analysis with a scRNAseq/TCRseq on 3 patients, two non-responders and one responder. Actually, this latter patient 1-006 is described as L198: "initial stable disease and a long survival" and as PD in Supp Table 2. More consistency in the data is required. Furthermore, this analysis is both too much (sophisticated and expensive analyses) and too little (only 3 patients, with mostly PD) to reach any statistical or biological significance.

Reviewer #3:

Remarks to the Author:

This is a very well written and clearly presented study that reports an interim analysis of a small (N=29) phase 2 clinical trial of the combination of Pembrolizumab (anti-PD1) and Entinostat (anti-HDAC) in uveal melanoma (UM) patients. The authors make a strong case (page 4) for trialing the addition of an HDAC inhibitor to the more usual regimen of PD1 inhibitor alone. There are some details that require clarification/expansion in order to more fully assess the relative benefits of this combined therapy over the standard anti-PD1 treatment.

The authors state that the primary endpoint was objective response rate (ORR), which was 14%, i.e. 4 patients. They argue this is better (though not statistically different) to the expected ORR of 5% observed in prior trials of anti-PD1 alone. Figure 1 includes plots of overall survival (OS) and progression free survival (PFS). The median values for OS and PFS should also be included in the Abstract and comments made on how they compare to trials of anti-PD1 alone.

The 4 patients showing an objective response (OR) comprised 3 with BAP1 wildtype tumors and a patient with an iris tumor. A priori, such patients are known to have better outcome regardless of treatment. It is therefore unclear whether the observed differences in OR in these patients is due to differences in underlying biology, or the drug treatment.

Similarly, there were several marked differences in baseline clinical status of the UM patients that may have also impacted the ORR. For example, 3 patients did not have liver metastasis (a good prognostic indicator) – were these the cases showing an OR? Were they the 3 BAP1 wildtype cases? Likewise, some patients had prior treatment with chemotherapy - were these the patients showing an OR? Did patients with lower AJCC metastasis stage show better OR? And was an OR associated with the number of doses of trial drugs? (it is stated that all patients got at least 1 dose but it is not clear how many got more than 1, or if multiple doses was associated with more

favorable response).

Adverse events (AEs) were reported in all but 1 patient, and two-thirds of patients had a severe (grade 3 or 4) AE. How do these values compare to anti-PD1 treatment alone? Comment should be made on the additional toxicity of adding the HDAC inhibitor to the treatment regimen.

The authors find that there was no correlation between TIL scores and survival or clinical benefit (page 7) – so what explains the 4 ORs? Is it simply differences in underlying biology or clinical status (see above) rather than drug treatment? The lack of a 'control arm' in this phase 2 trial is a major limitation in that no comparison can be made between the efficacy of anti-PD1 and anti-HDAC combination therapy, versus anti-PD1 treatment alone.

The last 2 lines of the Results state that "...engagement of the HDAC inhibitor can be measured as a specific gene expression changes in T cells.....". The evidence for this is extremely weak. Firstly, insufficient detail is given on how the 12-gene differential expression signature was derived. Secondly, 12 differentially expressed genes from the entire genome is less than that expected statistically by chance. There is no information (or validation study) to indicate the robustness of the 12-gene signature. I argue strongly that this aspect of the report be dropped.

Reviewer #1 (Remarks to the Author):

Ny, Jespersen et al present a very interesting Phase 2 trial of combined HDAC and PD-1 inhibition in uveal melanoma. There is a clear unmet need and the trial was well run. They have presented some interesting correlates suggesting that, among other things, non-BAP1 mutant UM may be responsive to this therapeutic approach. I have some suggestions that may improve the manuscript, below. Sincerely, Alex Shoushtari.

We thank Dr Shoushtari for his time reviewing this manuscript.

1) Safety – please elaborate on dose reductions/omissions and steroid or secondary immune suppression usage to give a better broader idea of safety and tolerability of the combo.

We have now elaborated on safety details, including the use of dose reductions/omissions and need for immunomodulatory agents in the results section.

2) Gene expression analysis appears well done, and 12 genes significant – but can you include a sentence on themes within those genes? This would introduce and better prepare the reader to understand the discussion focusing on these genes. Also, if improved antigen presentation was one of the purported mechanisms of synergy between HDAC inhibition and PD-1 blockade, do we have a hypothesis as to why no antigen presenting machinery or Class 1 HLA genes are in this subset (to the best of my ability to scroll through the supp table)?

We thank for this insightful comment, which we had not considered. This analysis only considers the state of gene expression before treatment, and it is not clear whether differences in baseline expression of antigen presentation genes between patients would associate with survival or not. We have not analyzed pre- and post-therapy biopsies so therefore we do not know if an induction or lack of induction of HLA/antigen presentation would correlate with survival. We can speculate that it would not, since our single cell sequencing data of pre- and post-therapy PBMC-derived cells show induction of HLA genes even though the three samples came from non-responders. Taken together, the results suggest that other factors are likely also involved in determining whether a given patient responds.

Regarding the 12 genes, this part has now been deleted from the manuscript based on the comments from the two other reviewers, specifically because their suggested relevance as potential biomarkers requires validation in an independent cohort.

Minor

1) P. 6 lines 173-74 – “responded to” what? PD-1 blockade?

We have changed this and thank Dr Shoushtari for finding this error.

2) When referring to dichotomous OS for PBMC analysis, “longest” and “shortest” imply a subset of extreme responders, whereas you are actually just comparing “longer” and “shorter” OS. Subtle but important distinction to better reflect the comparison.

We thank Dr Shoushtari for helping us avoiding writing what can easily be misinterpreted.

Reviewer #2 (Remarks to the Author):

In this manuscript, Ny and Colleagues are describing the PEMDAC phase 2 trial associating pembrolizumab and entinostat in patients with metastatic uveal melanoma (mUM). They provide their initial clinical findings, as well as exploratory biologic analyses. This manuscript contains a number of interesting data, well described. However the level of evidence is most often low, and it is difficult to draw any definitive conclusion from this report.

We thank the reviewer for taking the time to review our work and for providing insightful comments.

1) Clinical trail

The authors report the outcomes of 28 pts (one excluded from the study) with mUM treated in first line or not with Pembro/Entinostat in a single arm study. ORR was 14%, reaching primary endpoint, by with only PR (no CR). This ORR compares favorably with historical series of Pembro alone (ORR ~5%). However, the incidence and intensity of adverse events (AEs) was very high, especially when compared to historical series of Pembro, leading to the treatment discontinuation of 3 pts and 19 pts (66%) with \geq grade 3AEs (grade 3, 4 and 5 AEs corresponding to AEs ranging from “severe”, to “life-threatening” and “death related to AE”, respectively; the information about the precise grade should be given in this work).

We have now elaborated on safety details, including precise grade, the use of dose reductions-/omission and need for immunomodulatory agents in the results section.

In summary:

- Nine grade 3-4 irAEs were observed: hepatitis (N=4, all grade 3), skin toxicity (N=2, all grade 3), colitis (N=1, grade 3), stomatitis (N=1, grade 3) and hypophysitis (N=1, grade 4).
- Thirteen patients received immune modulating drugs for the management of irAE (corticosteroids monotherapy, N=12, and corticosteroids in combination with mycophenolate, N=1).
- Nine patients needed dose interruption and reduction of entinostat: neutropenia (with or without thrombocytopenia, N=5), nausea (N=3) and rash (N=1).
- Three patients had an AE leading to treatment discontinuation: pneumonitis (N=1, grade 2), hepatitis (N=1, grade 3), hypophysitis (N=1, grade 4).
- There were **no treatment-related deaths**.

As experienced medical oncologists treating patients with PD-1 inhibitors combined with e.g. other checkpoint inhibitors or targeted kinase inhibitors, we expected both the nature and severity of the side-effects and regarded them as manageable.

A paradox of mUM is the very long period of no or very mild clinical symptoms of the metastatic disease, in contrast with the very short survival. One may be reluctant to propose such drug combination for such limited benefice. Of note, classically in 3+3 phase I studies, the RP2D (recommended dose for phase II studies) is the dose producing ~20% of dose-limiting toxicity.

The phase 2 dose was shared to us by MSD and Syndax and was based on in-house phase 1 data from cutaneous melanoma. They did not include any patients with uveal melanoma which is why our study is important also from a safety perspective.

Our study is a single-arm Phase 2 study of a new combination and without a randomized trial we would be reluctant to suggest any treatment, including this. Unfortunately, a prospective trial on PD-1 inhibitors had not been motivated by retrospective analyses of off-label use. One of the reasons for doing this trial was therefore to offer an addition to PD-1 inhibitor to motivate the start of such a trial. At the time of database lock the trial data from combination ipilimumab-nivolumab had not been disclosed neither. Now we know that our trial data are not inferior to ip-nivo data but collectively, all data point to that new combinations are needed for patients with metastatic uveal melanoma.

Nevertheless, PR of 3 out of 4 BAP1wt patients is intriguing, counterintuitive (BAP1mut UM were predicted to be more “immune-responsive”) and quite interesting. However, this observation is a posteriori and needs to be confirmed on additional BAP1wt pts, a difficult trial to set-up.

We agree that the fact that BAP1 wildtype patients being the best responders does reduce the number of patients in follow up trials. One way forward is that hopefully our data motivates others to do mutational analyses of their trials using immunotherapy. To date we do not know if a) the *BAP1* wildtype status sensitized to the pembrolizumab, b) to entinostat or c) only to the combination. Our data (despite the small sample size) serves as a hypothesis that others can quickly address by assessing *BAP1* status in e.g. the ipi-nivo trials (Pelster et al.; Piulats et al.) or in the ongoing trials of HDAC inhibitor vorinostat in UM (NCT01587352). We have discussed this in the manuscript and thank the reviewer for emphasizing this.

One case is easily explained by the high mutation burden associated with the iris origin of the UM. In this case, we can suppose that Pembro would have been sufficient to obtain a tumor response. I would be interested in seeing images of tumors responding to this combo in other patients than the iris UM one.

It is entirely possible that iris tumors could also be responders to single-agent pembrolizumab treatment, which has not been reported. Our results suggest this as a worthwhile avenue for research in future clinical trials. We have added response images of additional responders as a new Supplementary Figure.

The observation that LDHlow ctDNAlow predicts longer OS is neither new, nor counterintuitive (less tumor mass, less aggressive disease, better survival).

We agree and we are content that liquid biopsies can be used also in uveal melanoma trials. Moreover, benchmarking of our locally developed methodology was important. We have analyzed our data more and our new results show that ctDNA levels correlate well with LDH and to some extent also associate more closely with the observed responses, suggesting that ctDNA measurements are useful for tracking progression in metastatic UM (new fig S1d, e, g).

Minor points:

L132: Levels of circulating tumor DNA corroborated the response findings. Actually, the ctDNA evolution looks more complex than that, and more analyses and comments should be implemented.

We thank the reviewer for this comment and agree we need to comment more. The sentence is exchanged for a more complete description of the data. We have also now finalized the

analysis of all patients after having recovered samples thought to be lost samples from one of the clinical sites. We have rerun the analyses (Fig. 1f-h) and the association between low ctDNA baseline levels and prolonged OS remains. While the study was under review we also analyzed the data in other ways, e.g. mutant reads per ml blood, but the complex pattern including spikes did not change. We do not know the nature of the complexity, but we cannot rule out technical measurement variations despite we have not seen this from patients with other diagnoses than uveal melanoma undergoing therapeutic response (eg Bjursten et al., 2019 Case Rep Oncol; Johansson et al., 2019 Biomol Detect Quantif). There is a possibility that we are observing cycling growth patterns under immune editing events, resulting in the escape of the tumor from immune surveillance.

Supp Fig 1. Please develop abbreviations (PD PR SD BOR,...) in the legends. The Journal is for general audience, not sole clinicians.

We thank the reviewer for this important point. We have explained these abbreviations in all legends.

Table 2. “Adverse events (AE) of grade ≥ 3 in all patients (n=29).” n is ambiguous. Please replace by “Adverse events (AE) of grade ≥ 3 (n=29) in all patients” or similar.

We thank the reviewer and have revised the title to reduce ambiguity.

2) Exploratory analyses

These explorations are quite large, but not reached any level of novelty or of significance: No correlation with outcome was found with PDL1 and TIL analyses. Genetic analyses identified an iris UM with high TMB, which was already reported. The clinical response of this type of patient (TMB is the same order than cutaneous melanoma) is not unusual under Pembro and the role of Entinostat in this response is not clear.

We were the first to describe a UV-induced TMB in iris melanoma (Karlsson et al., 2020 Nature Communications) and later it was verified in a larger dataset of primary uveal melanomas by the Hayward group (Johansson et al., 2020, Nature Communications). One of the outstanding questions in our papers was if a patient with iris melanoma would respond to immunotherapy or if the immune suppressive microenvironment of UM would preclude a response. We provide the first evidence of an iris melanoma patient responding to immunotherapy (albeit in combination with entinostat). We agree that high TMB could be expected to be associated with responses, but we found this observation of interest since it identifies a subset of UM patients that could benefit from immune checkpoint inhibitors, just like those with UM that have *MBD4* mutations. While these tumors are rare, the current approach of treating them as any other UM might potentially be a disservice to these patients. In addition, this finding lends further support to the notion that failures of immunotherapy in UM could partly be due to the generally low TMB.

The major finding is the enrichment of *BAP1*wt in responders. However, this provocative finding was done a posteriori finding and needs to be validated in an independent cohort for having significance.

We agree that we have made many hypothesis-generating findings that need further validation in archived material from trials (ipi-nivo) or ongoing trials (vorinostat). This is also suggested in the discussion as an avenue for future research. If the finding about *BAP1* status hold true

for further scrutiny they also serve as important biological questions as to why *BAP1* deficient tumors are more resistant and how this knowledge can be harnessed for development of new therapies.

Similarly, the predicted gene expression signature of 12 genes has just no scientific value if not validated in an independent cohort, and its report here is misleading, despite potential information it could contain.

We agree that without independent validation as potential biomarkers, the report and discussion of these genes might be considered too speculative at the present. Since gene expression is more difficult to analyze than mutation status of *BAP1* in prospective trials by others we have deleted these supplementary panels and the associated paragraph of the discussion section. Deleting this analysis does not impair the most important findings on clinical response rates, survival and their relations to *BAP1* mutational status and the finding from the patient with iris melanoma.

Blood analyses

This exploratory analysis has very little significance. The authors analyzed by multicolor FACS samples before and after treatment, measuring dozens of cell populations. The description of selected blood populations should consider this hidden multiple testing, and I predict the marginal differences observed between responders and non-responders will not hold appropriate and mandatory statistical corrections.

We thank the reviewer for this important point. We have now performed multiple testing analysis and adjusted the *p*-values. Some comparisons remain significant (monocytes with respect to short- and long-term survival, neutrophils and long-term survival), whereas others do not (lymphocytes and long-term survival, CD4⁺ T cell subsets). This has been clarified in the methods and legends.

The authors followed this analysis with a scRNAseq/TCRseq on 3 patients, two non-responders and one responder. Actually, this latter patient 1-006 is described as L198: “initial stable disease and a long survival” and as PD in Supp Table 2. More consistency in the data is required. Furthermore, this analysis is both too much (sophisticated and expensive analyses) and too little (only 3 patients, with mostly PD) to reach any statistical or biological significance.

All three patients were PD, but one had initial stable disease that then progressed. This has been clarified in this part of the text. While the analysis is certainly not the most cost-effective in the context of the amount of results obtained, the results presented are supported statistically and have biological significance. The data confirms at the single-cell level in blood that phenotypes of immune cells are systematically affected by the treatment, with alterations in the expression of checkpoint receptors as well as HLA gene expression, which suggests increased activation. We further show that this effect on gene expression and activation state is not just due to a shift in the proportions of different T cell clones that already exist in these states, but also occurs within individual T cell clones. This effect is observed despite the fact that these particular patients were PD, suggesting that the drug has a biological effect but that other mechanisms are active that prevent further responses in these (and probably other) patients.

Reviewer #3 (Remarks to the Author):

This is a very well written and clearly presented study that reports an interim analysis of a small (N=29) phase 2 clinical trial of the combination of Pembrolizumab (anti-PD1) and Entinostat (anti-HDAC) in uveal melanoma (UM) patients. The authors make a strong case (page 4) for trialing the addition of an HDAC inhibitor to the more usual regimen of PD1 inhibitor alone. There are some details that require clarification/expansion in order to more fully assess the relative benefits of this combined therapy over the standard anti-PD1 treatment.

We thank the reviewer for taking the time to review our work and for providing insightful comments.

The authors state that the primary endpoint was objective response rate (ORR), which was 14%, i.e. 4 patients. They argue this is better (though not statistically different) to the expected ORR of 5% observed in prior trials of anti-PD1 alone.

The observed ORR is significantly different from 5 % according to Simon two-stage design and applied significance-level. We refer to the Supplemental Method section for details.

Figure 1 includes plots of overall survival (OS) and progression free survival (PFS). The median values for OS and PFS should also be included in the Abstract and comments made on how they compare to trials of anti-PD1 alone.

We thank the reviewer for this suggestion. Median OS and PFS are now included in the abstract. To our knowledge, there are no published and completed prospective trials of PD-1 inhibitors alone in mUM, Comparison with published retrospective case series and benchmark data for previous trials is cautiously discussed in the discussion, focusing on OS.

The 4 patients showing an objective response (OR) comprised 3 with BAP1 wildtype tumors and a patient with an iris tumor. A priori, such patients are known to have better outcome regardless of treatment. It is therefore unclear whether the observed differences in OR in these patients is due to differences in underlying biology, or the drug treatment.

BAP1 loss is known to be associated with shorter survival as assessed from the time when the tumor was growing as a primary in the eye. It is not trivial that outcomes for such patients would also be worse after metastatic disease has been developed, and under combined epigenetic and immunotherapy. Here we demonstrate that BAP1 loss has continued significance also in a therapy setting during metastatic disease. We are not aware of any other study on metastatic uveal melanoma demonstrating that BAP1 wildtype tumors are more sensitive in general. The exact biological function of BAP1 as a uveal melanoma metastasis suppressor is not known. Similarly, no-one has, to our knowledge, reported outcomes for metastatic iris melanomas under any form of immunotherapy.

Similarly, there were several marked differences in baseline clinical status of the UM patients that may have also impacted the ORR. For example, 3 patients did not have liver metastasis (a good prognostic indicator) – were these the cases showing an OR? Were they the 3 BAP1 wildtype cases? Likewise, some patients had prior treatment with chemotherapy - were these the patients showing an OR?

No, the responding patients, two patients had liver metastasis only (*BAP1* wt), one had hepatic and extra hepatic disease (*BAP1* mutated iris melanoma), and one had extra hepatic disease only (*BAP1* wt). We found no association between previous chemo and response group, albeit previous chemotherapy treatment associated with longer OS.

Did patients with lower AJCC metastasis stage show better OR? And was an OR associated with the number of doses of trial drugs? (it is stated that all patients got at least 1 dose but it is not clear how many got more than 1, or if multiple doses was associated with more favorable response).

Sixty percent of patients were M1a and 3 of 4 of responders were M1a. With such a small sample size and few responders, it is not possible to gain significance in that analysis. As seen in figure 1c, two responses were seen at the first evaluation, and two occurred after more than 6 months of therapy. In other words, number of doses did not associate with response.

Adverse events (AEs) were reported in all but 1 patient, and two-thirds of patients had a severe (grade 3 or 4) AE. How do these values compare to anti-PD1 treatment alone? Comment should be made on the additional toxicity of adding the HDAC inhibitor to the treatment regimen.

The experienced medical oncologists of the trial are used to treating patients with PD-1 inhibitors combined with e.g. other checkpoint inhibitors or targeted kinase inhibitors. Therefore, they were not unfamiliar with the nature and severity of the side-effects and regarded them as manageable. Entinostat toxicity was primarily neutropenia and thrombocytopenia but an exacerbation of the PD-1 inhibitor toxicity is also likely. We have now elaborated on safety details, including precise grade, the use of dose reductions-/omission and need for immunomodulatory agents in the results section.

In summary:

- Nine grade 3-4 irAEs were observed: hepatitis (N=4, all grade 3), skin toxicity (N=2, all grade 3), colitis (N=1, grade 3), stomatitis (N=1, grade 3) and hypophysitis (N=1, grade 4).
- Thirteen patients received immune modulating drugs for the management of irAE (corticosteroids monotherapy, N=12, and corticosteroids in combination with mycophenolate, N=1).
- Nine patients needed dose interruption and reduction of entinostat: neutropenia (with or without thrombocytopenia, N=5), nausea (N=3) and rash (N=1).
- Three patients had an AE leading to treatment discontinuation: pneumonitis (N=1, grade 2), hepatitis (N=1, grade 3), hypophysitis (N=1, grade 4).
- There were **no treatment-related deaths**.

The authors find that here was no correlation between TIL scores and survival or clinical benefit (page 7) – so what explains the 4 ORs? Is it simply differences in underlying biology or clinical status (see above) rather than drug treatment? The lack of a ‘control arm’ in this phase 2 trial is a major limitation in that no comparison can be made between the efficacy of anti-PD1 and anti-HDAC combination therapy, versus anti-PD1 treatment alone.

We agree with the reviewer that a lack of a control arm is a limitation but when designing the trial initially we were aware of the lack of effect of single PD-1 inhibitor and therefore wished to do a traditional phase 2 trial to see if there was significant benefit to add an HDAC

inhibitor as we had seen in preclinical models. We can conclude that this was unfortunately not the outcome of the trial, but in keeping with the original protocol we still analyzed exploratory endpoints such as genetics, immunology and liquid biopsies in the trial. These revealed correlations between *BAP1* status/tumor burden (ctDNA and LDH) and survival/responses. Whereas the liquid biopsy data has been demonstrated very recently in trials for patients with different cancer diagnoses, our trial data ought to be of interest to the community of clinicians dealing with patients with UM. Furthermore, *BAP1* status and an UV signature in an iris melanoma informs for other trialists that genetics can be very informative.

The last 2 lines of the Results state that "...engagement of the HDAC inhibitor can be measured as a specific gene expression changes in T cells.....". The evidence for this is extremely weak. Firstly, insufficient detail is given on how the 12-gene differential expression signature was derived. Secondly, 12 differentially expressed genes from the entire genome is less than that expected statistically by chance. There is no information (or validation study) to indicate the robustness of the 12-gene signature. I argue strongly that this aspect of the report be dropped.

We performed statistical analyses of gene expression in pre-treatment biopsies from patients with longer or shorter survival in the trial. We performed multiple testing corrections and the data are adjusted p-values (q). After that analysis only 12 genes remained. We have deleted the data since it was challenged by reviewer 2 and we will not be able to validate these genes in an independent cohort. Nevertheless, deleting that analysis does not impair the most important findings on clinical response rates, survival and their relation to *BAP1* mutational status and the finding from the patient with iris melanoma.

Regarding the scSeq data: The genes selected in this analysis was not based on the 12-gene signature that is now deleted, but rather on differences in the expression of genes in PBMC isolates before and after treatment. In addition, we find subsets of these genes altered also in uveal melanoma cell lines treated with entinostat (Sah et al., BiorXiv, 2021). The subset of genes shown in Fig. 3d represent some of these genes, that have known functions of the genes, eg in exhaustion of T cells. The results presented are supported statistically and have biological significance. The data confirms at the single-cell level in blood that phenotypes of immune cells are systematically affected by the treatment, with alterations in the expression of checkpoint receptors as well as HLA gene expression, which suggests increased activation. We further show that this effect on gene expression and activation states is not just due to a shift in the proportions of different T cell clones that already exist in these states, but also occurs within individual T cell clones. This effect is observed despite the fact that these particular patients were PD, suggesting that the drug has a biological effect here but that other mechanisms are active that prevent further responses in these (and probably other) patients. We have toned down the scSeq data by moving some of it to Supplemental Figures. The data do not distract from the most important findings of the manuscript but are descriptive evidence of an effect on gene expression in blood cells of patients in the PEMDAC trial.

Reviewers' Comments:

Reviewer #1:

Remarks to the Author:

The authors have addressed my relatively minor concerns.

Reviewer #2:

Remarks to the Author:

The authors have answered most of the questions addressed for their first submission and have well improved the quality of the manuscript, except few typos to be corrected:

L53 (Abstract): "low baseline ctDNA levels or LDH, and a 12-gene signature of mostly HDAC..."

Please remove the 12-gene signature that you removed in the main text.

L144: "LHD" please correct for LDH.

L181 "and the tumor suppressor gene BAP1, which coincided with 3q monosomy". I guess the authors mean "and the tumor suppressor gene BAP1, which coincided with 3p monosomy" or "chr 3 monosomy".

Table 1: Please uniform abbreviation for LDH (LD in the Table LDH in the text)

Spacing of text in Figures appears uneven. Please check in final Figures.

Reviewer #3:

Remarks to the Author:

The authors have done a highly commendable job of responding to each of the reviewer suggestions and concerns. The additional information now provided, particularly with regard to adverse events and dose reductions or omissions, gives a more thorough summary of the clinical aspects of patient responses in the trial. The additional data given in the rebuttal also give greater clarity on genetic and pathological correlations for the patients and serve to rule out more trivial explanations for the observed associations with response. The findings of the study establish a few hypotheses that hopefully will be tested at some stage by these authors and other clinicians working in this field.

Rebuttal Ny/Jespersen et al.,

Editorial

We were asked to report all secondary endpoints of our trial. We have added in a list of secondary endpoints in Supplementary Information containing information were in the manuscript or figures the secondary endpoint data can be found. We also added in the quality of life assessments as a new Supplementary figure 2 a-b.

Reviewer #1 (Remarks to the Author):

The authors have addressed my relatively minor concerns.

We thank Dr Shoushtari for his time reviewing this manuscript.

Reviewer #2 (Remarks to the Author):

The authors have answered most of the questions addressed for their first submission and have well improved the quality of the manuscript, except few typos to be corrected: L53 (Abstract): “low baseline ctDNA levels or LDH, and a 12-gene signature of mostly HDAC...” Please remove the 12-gene signature that you removed in the main text.

We thank the reviewer for taking the time to review our work and for identifying these typos. We have corrected this.

L144: “LHD” please correct for LDH.

We have corrected this.

L181 “and the tumor suppressor gene BAP1, which coincided with 3q monosomy”. I guess the authors mean “and the tumor suppressor gene BAP1, which coincided with 3p monosomy” or ‘chr 3 monosomy”

We agree and have corrected this.

Table 1: Please uniform abbreviation for LDH (LD in the Table LDH in the text)

We agree and have corrected this.

Spacing of text in Figures appears uneven. Please check in final Figures.

This was caused by a combability problem between our version of Illustrator and the pdf conversion in the Editorial Management System. We have fixed this now.

Reviewer #3 (Remarks to the Author):

The authors have done a highly commendable job of responding to each of the reviewer suggestions and concerns. The additional information now provided, particularly with regard to adverse events and dose reductions or omissions, gives a more thorough summary of the clinical aspects of patient responses in the trial. The additional data given in the rebuttal also

give greater clarity on genetic and pathological correlations for the patients and serve to rule out more trivial explanations for the observed associations with response. The findings of the study establish a few hypotheses that hopefully will be tested at some stage by these authors and other clinicians working in this field.

We thank the reviewer for the kind words and for taking the time to review our work.

Reviewers' Comments:

Reviewer #1:

Remarks to the Author:

The authors have addressed my critiques and I enjoyed reading this manuscript.